# Once is Enough: A Lightweight Cross-Attention for Fast Sentence Pair Modeling

**Yuanhang Yang**[1]  **Shiyi Qi**[1]  **Chuanyi Liu**[1]  **Qifan Wang**[2]
**Cuiyun Gao**[1]  **Zenglin Xu**[1]

[1]Harbin Institute of Technology, Shenzhen, China
[2]Meta AI, CA, USA

{ysngkil, syqi12138}@gmail.com   liuchuanyi@hit.edu.cn   wqfcr@fb.com
{gaocuiyun, xuzenglin}@hit.edu.cn

## Abstract

Transformer-based models have achieved great success on sentence pair modeling tasks, such as answer selection and natural language inference (NLI). These models generally perform *cross-attention* over input pairs, leading to prohibitive computational costs. Recent studies propose dual-encoder and late interaction architectures for faster computation. However, the balance between the expressive of cross-attention and computation speedup still needs better coordinated. To this end, this paper introduces a novel paradigm *MixEncoder* for efficient sentence pair modeling. MixEncoder involves a lightweight cross-attention mechanism. It avoids the repeated encoding of the same query for different candidates, thus allowing modeling the query-candidate interaction in parallel. Extensive experiments conducted on four tasks demonstrate that our MixEncoder can speed up sentence pairing by over 113x while achieving comparable performance as the more expensive cross-attention models. The source code is available at `https://github.com/ysngki/MixEncoder`.

## 1 Introduction

Sentence pair modeling, such as natural language inference, question answering, and information retrieval, is an essential task in natural language processing (Nogueira and Cho, 2020; Qu et al., 2021; Zhao et al., 2021). These tasks can be depicted as a procedure of scoring the candidates given a query. Recently, Transformer-based models (Vaswani et al., 2017; Devlin et al., 2019) have shown promising performance on sentence pair modeling tasks due to the expressiveness of the pre-trained cross-encoder. As shown in Figure 1(a), the cross-encoder takes a pair of query and candidate as input and calculates the interaction between them at each layer by the input-wide self-attention mechanism. Despite the effective text representation power, the cross-encoder leads to exhaustive

computation costs, especially when the number of candidates is very large ( e.g., the interaction will be calculated $N$ times if there are $N$ candidates). This computation cost, therefore, restricts the use of these cross-encoder models in many real-world applications (Chen et al., 2020).

To tackle this issue, we propose a lightweight cross-attention mechanism, called MixEncoder, that speeds up the inference while maintaining the expressiveness of cross-attention. Specifically, the proposed MixEncoder accelerates the cross-attention by performing attention only from candidates to the query, involving few tokens and only at a few layers. This lightweight cross-attention avoids repetitive query encoding, supporting the processing of multiple candidates in parallel and thus reducing computation costs. Additionally, MixEncoder allows to pre-compute the candidates into several dense context embeddings and to store them offline to accelerate the inference further.

We evaluate MixEncoder for sentence pair modeling on four benchmark datasets related to tasks of natural language inference, dialogue, and information retrieval. The results demonstrate that MixEncoder better balances the effectiveness and efficiency. For example, MixEncoder achieves a substantial speedup of more than 113x over the cross-encoder and provides competitive performance.

## 2 Background

Extensive studies, including dual-encoder (Reimers and Gurevych, 2019) and *late interaction* models (MacAvaney et al., 2020; Gao et al., 2020; Chen et al., 2020; Khattab and Zaharia, 2020), have been proposed to accelerate the transformer inference on sentence pair modeling tasks.

As shown in Figure 1, dual-encoders process the query and candidates separately, allowing pre-computing the candidates to accelerate online inference, resulting in fast inference speed. However, this speedup is built upon sacrificing the expres-

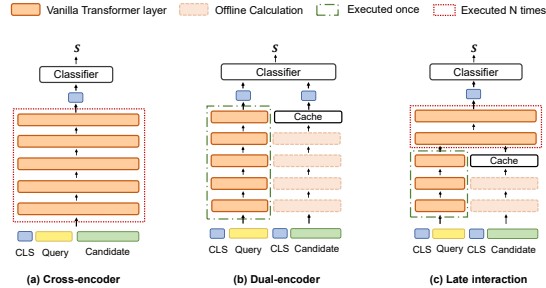

Figure 1: Illustration of three popular sentence pair approaches, where $N$ denotes the number of candidates and $s$ denotes the relevance score of candidate-query pairs. The cache stores the pre-computed embeddings.

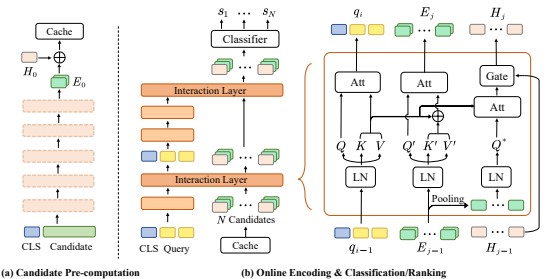

Figure 2: Overview of proposed MixEncoder.

siveness of cross-attention (Luan et al., 2021; Hu et al., 2021; Zhang et al., 2021). Alternatively, late-interaction models adjust dual-encoders by appending an interaction component, such as a stack of Transformer layers (Cao et al., 2020; Nie et al., 2020), for modeling the interaction between the query and the cached candidates. These approaches still suffer from the high costs of the interaction component (Chen et al., 2020).

## 3 Method

In this section, we introduce the details of the proposed MixEncoder, which simplifies cross-attention by enabling pre-computation, reducing the times of query encoding, and reducing the number of involved tokens and layers.

### 3.1 Candidate Pre-computation

Given a candidate that is a sequence of tokens $T_i = [t_1, \cdots, t_l]$, we experiment with two strategies to encode these tokens into $k$ context embeddings in advance, where $k \ll l$: (1) prepending $k$ special tokens $\{S_i\}_{i=1}^k$ to $T_i$ before feeding $T_i$ into the Transformer encoder (Vaswani et al., 2017; Devlin et al., 2019), and using the output at these special tokens as context embeddings ($S$-strategy); (2) maintaining $k$ context codes (Humeau et al.,

2020) to extract global features from output of the encoder by attention mechanism ($C$-strategy). The default configuration is $S$-strategy as it provides slightly better performance. The pre-computed context embeddings $E \in \mathbb{R}^{N \times k \times d}$ are cached for online inference, where $N$ is the number of candidates.

#### 3.1.1 Query Encoding

Since the cross-encoder performs $N$ times of query encoding, which contributes to the inefficiency, a straightforward way to accelerate the inference is to reduce the encoding times of the query. Here we encode the query without taking its candidates into account, thus requiring the encoding only once.

To preserve the expressiveness of the cross-attention, the simplified cross-attention is performed at several *interaction layers*. As shown in Figure 2, the context embeddings $E_{j-1}$ of candidates are allowed to attend over the intermediate token embeddings of the query, thus obtaining context-aware representations $E_j$ and $H_j$ for the query and its candidates.

Concretely, at each interaction layer, the key and value matrices of the query are utilized by candidates in two ways. (1) Producing contextualized representations for the candidates:

$$E_j = \text{Attn}(Q', [K'; K], [V'; V]), \qquad (1)$$

where $Q'$, $K'$, $V'$ are derived from the $E_{j-1}$ with a linear transformation. $E_j$ is supposed to contain semantics from both the query and candidates. (2) Compressing the semantics of the query into a vector for each candidate:

$$H_j = \text{Gate}(\text{Attn}(Q^*, K, V), H_{j-1}), \qquad (2)$$

where $Q^* \in \mathbb{R}^{N \times d}$ is derived from $E_{j-1}$ by a pooling operation, $H \in \mathbb{R}^{N \times d}$ stands for the candidate-aware query states and $H_0$ is initialized as a zero matrix.

### 3.2 Prediction

Let $H$ and $E$ denote the query states and the candidate context embeddings generated by the last interaction layer, respectively. For the $i$-th candidate, its representation is the mean of the $i$-th row of $E$, denoted as $e_i$. The representation of the query with respect to this candidate is the $i$-th row of $H$, denoted as $h_i$. The cosine similarity between $e_i$ and $h_i$ is used as the semantic similarity. Additionally, we can pass $e_i$ and $h_i$ to a classifier for classification tasks.

Table 1: Time Complexity of the attention module. We use $q$, $c$ to denote the query and candidate length, respectively. $d$ indicates the hidden layer dimension, $N$ indicates the number of candidates for each query and $k$ indicates the number of context embeddings for each candidate.

| Model | Total ($N = 1$) | Pre-computation ($N = 1$) | Online |
|---|---|---|---|
| Dual-BERT | $d(c^2 + q^2) + d^2(c + q)$ | $dc^2 + d^2c$ | $dq^2 + d^2q$ |
| Cross-BERT | $d(c + q)^2 + d^2(c + q)$ | $0$ | $N(d(q + c)^2 + d^2(q + c))$ |
| MixEncoder | $d(c^2 + q(q + k) + k^2) + d^2(c + q + k)$ | $dc^2 + d^2c$ | $dq^2 + d^2q + N(k + q + d)dk$ |

## 3.3 Time Complexity

Table 1 presents the time complexity of the Dual-BERT, Cross-BERT, and our proposed MixEncoder. We can observe that MixEncoder supports offline pre-computation to reduce the online time complexity. During the online inference, the query encoding cost term $(dq^2 + d^2q)$ of MixEncoder does not increase with the number of candidates since it conducts query encoding only once. Moreover, MixEncoder's query-candidate term $N(k + q + d)dk$ can be reduced by setting $k$ as a small value, which can further speed up the inference.

## 4 Experiments

**Datasets.** We evaluate MixEncoder on three paired-input tasks over four datasets, including MNLI (Williams et al., 2018) for natural language inference, MS MARCO passage reranking (Bajaj et al., 2018) for information retrieval, and DSTC7 (Yoshino et al., 2019), Ubuntu V2 (Lowe et al., 2015) for utterance selection for dialogue.

**Baselines.** (1) Cross-BERT is the original BERT (Devlin et al., 2019). (2) Dual-BERT (Sentence-BERT) is proposed by Reimers et al. (Reimers and Gurevych, 2019). (3) Deformer (Cao et al., 2020) is a decomposed Transformer that utilizes lower layers to encode sentences separately and then uses upper layers to encode text pairs together. (4) Poly-Encoder (Humeau et al., 2020) encodes the query and its candidates separately and performs a light-weight late interaction. (5) ColBERT (Khattab and Zaharia, 2020) is a late interaction model which adopts the MaxSim operation to obtain relevance scores. This operation prohibits the utilization of ColBERT on classification tasks. (6) VIRT (Li et al., 2022) performs the cross-attention at the last layer and utilizes knowledge distillation during training.

**Training Details.** While training models on MNLI, we use the labels provided in the dataset. While training models on the other three datasets, we use in-batch negatives (Karpukhin et al., 2020; Qu et al., 2021). Detailed settings are provided in A.1.

## 5 Results

Table 2 shows the experimental results of baselines and three variants of MixEncoder. We measure the inference time of all the baseline models for queries with 1000 candidates and report the speedup.

### 5.1 Performance Comparison

**Variants of MixEncoder.** To study the effect of the number of interaction layers and that of the number of context embeddings per candidate, we consider three variants, denoted as MixEncoder-a, -b, and -c, respectively. Specifically, MixEncoder-a and -b set $k$ as 1. The former performs interaction at the last layer and the latter performs interaction at the last three layers. MixEncoder-c is similar to MixEncoder-b but with $k = 2$.

**Dual-BERT and Cross-BERT.** The performance of the dual-BERT and cross-BERT are reported in the first two rows of Table 2. We can observe that MixEncoder consistently outperforms the Dual-BERT. The variants with more interaction layers or more context embeddings generally yield more improvement. For example, on DSTC7, MixEncoder-a and MixEncoder-b achieve an improvement by $0.7\%$ (absolute) and $1.6\%$ over the Dual-BERT, respectively. Moreover, MixEncoder-a provides comparable performance to the Cross-BERT on both Ubuntu and DSTC7. MixEncoder-b can even outperform the Cross-BERT on DSTC7 ($+0.6$), since MixEncoder can benefit from a large batch size (Humeau et al., 2020). However, the effectiveness of the MixEncoder on MS MARCO is slight.

We can find that the difference in the inference time between the Dual-BERT and MixEncoder is minimal, while Cross-BERT is 2 orders of magnitude slower than these models.

Table 2: Performance of Dual-BERT, Cross-BERT and three variants of MixEncoder on four datasets.

| Model | MNLI Accuracy | Ubuntu R1@10 | Ubuntu MRR | DSTC7 R1@100 | DSTC7 MRR | MS MARCO R1@1000 | MS MARCO MRR(dev) | Speedup Times | Space GB |
|---|---|---|---|---|---|---|---|---|---|
| Cross-BERT | $83.7_{0.1}$ | $83.1_{0.7}$ | $89.4_{0.5}$ | $66.8_{0.6}$ | $75.2_{0.4}$ | **23.3** | **36.0** | 1.0x | - |
| Dual-BERT | $75.2_{0.1}$ | $81.6_{0.2}$ | $88.5_{0.1}$ | $65.8_{1.0}$ | $74.2_{0.7}$ | 20.3 | 32.2 | **132x** | 0.3 |
| PolyEncoder-64 | $76.8_{0.1}$ | $82.3_{0.5}$ | $88.9_{0.4}$ | $66.4_{1.5}$ | $74.8_{0.9}$ | 20.3 | 32.3 | 130x | 0.3 |
| PolyEncoder-360 | $77.3_{0.2}$ | $81.8_{0.2}$ | $88.6_{0.1}$ | $65.7_{0.6}$ | $74.0_{0.3}$ | 20.5 | 32.4 | 127x | 0.3 |
| ColBERT | $\times$ | $82.9_{0.3}$ | $89.3_{0.2}$ | $67.2_{0.7}$ | $74.8_{0.4}$ | 22.8 | 35.4 | 35.2x | 8.6 |
| VIRT | $78.3_{0.3}$ | $83.1_{0.2}$ | $89.4_{0.2}$ | $66.5_{0.7}$ | $74.9_{0.2}$ | 21.5 | 32.3 | 33.3x | 52.7 |
| Deformer | $82.0_{0.1}$ | $83.2_{0.4}$ | $89.5_{0.2}$ | $66.3_{1.0}$ | $75.3_{0.6}$ | 23.0 | 35.7 | 1.9x | 52.7 |
| MixEncoder-a | $77.5_{0.4}$ | $83.1_{0.1}$ | $89.4_{0.1}$ | $66.9_{0.5}$ | $74.9_{0.2}$ | 20.4 | 32.0 | 113x | 0.3 |
| MixEncoder-b | $77.8_{0.2}$ | $83.2_{0.0}$ | $89.5_{0.1}$ | $68.2_{0.8}$ | $75.8_{0.5}$ | 20.7 | 32.5 | 89.6x | 0.3 |
| MixEncoder-c | $78.4_{0.4}$ | $83.3_{0.1}$ | $89.5_{0.0}$ | $66.7_{0.4}$ | $74.8_{0.3}$ | 20.0 | 31.9 | 84.8x | 0.6 |

Table 3: Ablation analysis for MixEncoder-a and -b.

| Variants | Ubuntu -a | Ubuntu -b | DTSC7 -a | DTSC7 -b |
|---|---|---|---|---|
| Original | **89.5** | **89.5** | **74.9** | **76.1** |
| w/o $H$ | 88.9 | 89.1 | 74.0 | 73.9 |
| w/o $E$ | 89.2 | 89.3 | 74.8 | 75.2 |

**Late Interaction Models.** From Table 2, we have the following observations. First, among all the late interaction models, Deformer that adopts a stack of Transformer layers as the late interaction component consistently shows the best performance on all the datasets. This demonstrates the effectiveness of cross-attention. In exchange, Deformer shows limited speedup (1.9x). Compared to the ColBERT and Poly-Encoder, MixEncoder outperforms them on the datasets except for MS MARCO. Although ColBERT consumes more computation than MixEncoder, it shows worse performance than MixEncoder on DSTC7 and Ubuntu. This demonstrates that the lightweight cross-attention can achieve a better trade-off between efficiency and effectiveness. However, on MS MARCO, MixEncoder and poly-encoder lag behind the ColBERT by a large margin. We conjecture that MixEncoder falls short of handling term-level matching. We will elaborate on it in section A.4.

## 5.2 Ablation Study

**Representations.** We conduct ablation studies to quantify the impact of two key components ($E$ and $H$) utilized in MixEncoder. The results are shown in Table 3. All components contribute to a gain in performance. It demonstrates that the simplified cross-attention can produce effective representations for both the query and its candidates.

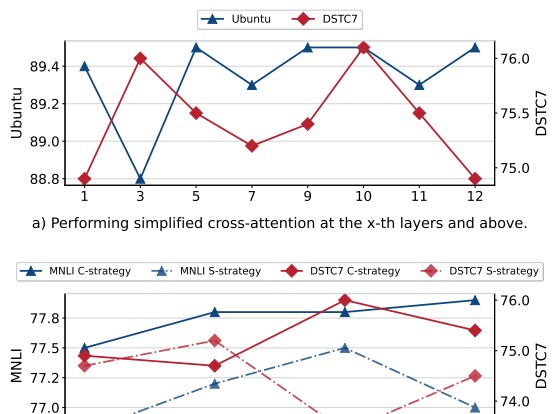

a) Performing simplified cross-attention at the x-th layers and above.

b) Model with a varing number of context vectors and different strategies.

Figure 3: Parameter analysis on the interaction layers and pre-computed context embeddings.

**Interaction layers.** Figure 3(a) shows the results when MixEncoder performs interaction at Transformer layers upper than $x$. Increasing interaction layers cannot continuously improve the ranking quality. On both Ubuntu and DSTC7, the performance of MixEncoder achieves a peak with the last three layers utilized for interaction. More experiments are reported in section A.6.

**Context embeddings.** We study the effect of the number of candidate embeddings and the pre-computation strategies with the last layer to perform the simplified cross-attention. From Figure 3(b), it is observed that the $S$-strategy generally outperforms the C-strategy, and a larger $k$ can lead to a better performance for the $S$-strategy.

Table 4 shows the average time per example for different models. It is shown that MixEncoder consumes more time as $k$ increases. Nevertheless, the difference in timing between Dual-BERT and MixEncoder is rather minimal, whereas Cross-BERT

is significantly slower by two orders of magnitude.

Table 4: Query processing times with 1,000 candidates and the last layer utilizing simplified cross-attention.

| Model | Time (ms) |
| --- | --- |
| Dual-BERT | 7.2 |
| Cross-BERT | 949.4 |
| MixEncoder (k=1) | 8.4 |
| MixEncoder (k=2) | 9.1 |
| MixEncoder (k=3) | 10.0 |
| MixEncoder (k=4) | 11.5 |
| MixEncoder (k=10) | 24.3 |

## 6 Conclusion

In this paper, we propose MixEncoder to balance the trade-off between performance and efficiency. It involves a lightweight cross-attention mechanism that allows us to encode the query once and process all the candidates in parallel. Experimental results demonstrate that MixEncoder can speed up sentence pairing by over 113x while achieving comparable performance as the more expensive cross-attention models.

## 7 Acknowledgements

This work was partially supported by the National Key Research and Development Program of China (No. 2018AAA0100204), a key program of fundamental research from Shenzhen Science and Technology Innovation Commission (No. JCYJ20200109113403826), the Major Key Project of PCL (No. 2022ZD0115301), and an Open Research Project of Zhejiang Lab (NO.2022RC0AB04).

## Limitations

Although MixEncoder has been demonstrated to be effective in cross-attention computation, we recognize that MixEncoder does not perform well on MS MARCO. It indicates that our MixEncoder falls short of detecting token overlapping since it loses token-level features by pre-encode candidates into several context embeddings. Moreover, MixEncoder is not evaluated on a large-scale evaluation dataset, such as an end-to-end retrieval task, which requires the model to retrieve top-$k$ candidates from millions of candidates (Qu et al., 2021; Khattab and Zaharia, 2020).

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

## A More Details

### A.1 Training Details

For Cross-BERT and Deformer, which require exhaustive computation, we set the batch size as 16 due to the limitation of computation resources. For other models, we set the batch size as 64. All the models use BERT (based, uncased) with 12 layers and fine-tune it for up to 50 epochs with a learning rate of 1e-5 and linear scheduling. All experiments are conducted on a server with 4 Nvidia Tesla A100 GPUs, which have 40 GB graphic memory.

### A.2 Datasets

The statistics of datasets are detailed in Table 5. We use accuracy to evaluate the classification performance on MNLI. For other datasets, MRR and recall are used as evaluation metrics.

Table 5: Statistics of experimental datasets.

|  | Dataset | MNLI | MS MACRO | DSTC7 | Ubuntu V2 |
|---|---|---|---|---|---|
| Train | # of queries | 392,702 | 498,970 | 200,910 | 500,000 |
|  | Avg length of queries | 27 | 9 | 153 | 139 |
|  | Avg length of candidates | 14 | 76 | 20 | 31 |
| Test | # of queries | 9,796 | 6,898 | 1,000 | 50,000 |
|  | # of candidates per query | 1 | 1000 | 100 | 10 |
|  | Avg length of queries | 26 | 9 | 137 | 139 |
|  | Avg length of candidates | 14 | 74 | 20 | 31 |

### A.3 In-batch Negative Training

We change the batch size and show the results in Figure 4. It can be observed that increasing batch size contributes to better performance. Moreover, we have the observation that models may fail to diverge with small batch sizes. Due to the limitation of computation resources, we set the batch size as 64 for our training.

### A.4 Error Analysis

In this section, we take a sample from MS MARCO to analyze our errors. We observe that MixEncoder falls short of detecting token overlapping. Given the query "foods and supplements to lower blood sugar", MixEncoder fails to pay attention to the keyword "supplements," which appears in both the query and the positive candidate. We conjecture that this drawback is due to the pre-computation that represents each candidate into $k$ context embeddings. It loses the token-level features of the candidates. On the contrary, ColBERT caches all the token embeddings of the candidates and estimates relevance scores based on token-level similarity.

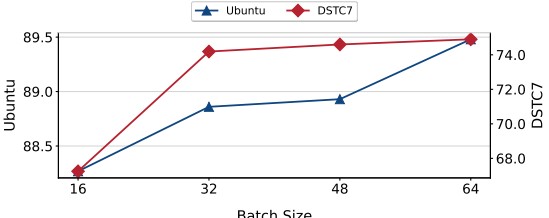

Figure 4: Parameter analysis on the batch size.

### A.5 Inference Speed

We conduct speed experiments to measure the online inference speed for all the baselines. Concretely, we sample 100 samples from MS MARCO. Each of the samples has roughly 1000 candidates. We measure the time for computations on the GPU and exclude time for text reprocessing and moving data to the GPU.

Table 6: Time to evaluate 100 queries with 1k candidates. The Space used to cache the pre-computed embeddings for 1k candidates are shown.

| Model | Time (ms) 1k | Space (GB) 1k |
|---|---|---|
| Dual-BERT | 7.2 | 0.3 |
| PolyEncoder-64 | 7.3 | 0.3 |
| PolyEncoder-360 | 7.5 | 0.3 |
| ColBERT | 27.0 | 8.6 |
| Deformer | 488.7 | 52.7 |
| Cross-BERT | 949.4 | - |
| MixEncoder-a | 8.4 | 0.3 |
| MixEncoder-b | 10.6 | 0.3 |
| MixEncoder-c | 11.2 | 0.6 |

### A.6 Interaction Layers

From Table 7, it is observed that performing cross-attention at higher layers generally yields better performance. Since we use the output of the final interaction layers as the sentence embeddings, choosing low layers enables the early exit mechanism.

Table 7: Results (Recall@1) of performing simplified cross-attention at two interaction layers on DSTC.

| Layer | 12 | 10 | 8 |
|---|---|---|---|
| 2 | 65.4 | 64.8 | 64.3 |
| 4 | 66.4 | 65.7 | 66.2 |
| 6 | 67.1 | 65.5 | 66.0 |
| 8 | 66.6 | 65.4 | - |
| 10 | 67.4 | - | - |