# OpenReview forum: "Once is Enough: A Light-Weight Cross-Attention for Fast Sentence Pair Modeling"
_EMNLP/2023/Conference — EMNLP 2023 Main_

### Official Review · Reviewer_go6q · 2023-08-03

**Soundness:** 3

**Excitement:**

3: Ambivalent: It has merits (e.g., it reports state-of-the-art results, the idea is nice), but there are key weaknesses (e.g., it describes incremental work), and it can significantly benefit from another round of revision. However, I won't object to accepting it if my co-reviewers champion it.

**Paper Topic And Main Contributions:**

In this paper, the authors propose MixEncoder to balance the trade-off between performance and efficiency. MixEncoder uses a lightweight cross-attention mechanism to encode the query once and process all the candidates in parallel. Experimental results demonstrate that MixEncoder can speed up sentence pairing by over 113x while achieving comparable performance as the more expensive cross-attention models.

**Questions For The Authors:**

MixEncoder only considers BERT-base and does not do experiments on large language models to verify its effectiveness. How does MixEncoder perform on large language models?

**Reasons To Accept:**

1. This paper proposes MixEncoder, which simplifies cross-attention by enabling pre-computation, reducing the times of query encoding, and reducing the number of involved tokens and layers.
2. This paper conducts lots of experiments.

**Reasons To Reject:**

1. MixEncoder only considers BERT-base and does not do experiments on large language models to verify its effectiveness.
2. The references need to be cleaned up.

**Reproducibility:**

4: Could mostly reproduce the results, but there may be some variation because of sample variance or minor variations in their interpretation of the protocol or method.

**Reviewer Confidence:**

4: Quite sure. I tried to check the important points carefully. It's unlikely, though conceivable, that I missed something that should affect my ratings.

---

> ### Author Rebuttal · Authors · 2023-08-29
>
> We sincerely appreciate your insightful comments
>
> **Re: "MixEncoder only considers BERT-base and does not do experiments on large language models to verify its effectiveness."**
>
> Thank you for this feedback. Following [1] , our research primarily focuses on the application of BERT-base in our experiments. We agree that experiments on large language models may benefit our work. To this end, we consider a larger variant: BERT-large. Due to time constraints, the additional experiments are conducted on DSTC7. The table below reports the results. It is found that MixEncoder continues to enhance the performance of Dual-BERT even after the model size is scaled up. We plan to incorporate this table into the appendix for further reference.
>
> | Method       | BERT-base    | BERT-large   |
> | ------------ | ------------ | ------------ |
> | Cross-BERT   | $75.2_{0.5}$ | $77.0_{0.2}$ |
> | Dual-BERT    | $74.2_{0.5}$ | $75.9_{0.4}$ |
> | MixEncoder-a | $74.9_{0.2}$ | $76.3_{0.4}$ |
> | MixEncoder-b | $75.8_{0.5}$ | $76.5_{0.3}$ |
>
> [1] Humeau, S., Shuster, K., Lachaux, M., & Weston, J. (2019). Poly-encoders: Architectures and Pre-training Strategies for Fast and Accurate Multi-sentence Scoring. *International Conference on Learning Representations*.
>
>
>
> **Re: "The references need to be cleaned up."**
>
> We appreciate your valuable suggestions. We will clean up the references to enhance the quality of our paper.

---

### Official Review · Reviewer_TknS · 2023-08-04

**Soundness:** 4

**Excitement:**

4: Strong: This paper deepens the understanding of some phenomenon or lowers the barriers to an existing research direction.

**Paper Topic And Main Contributions:**

This paper is about accelerating cross-attention mechanism in the scenario of sentence pairing. The main contribution is to propose a lightweight paradigm MixEncoder to speed up by over 113x and maintaining good performance for sentence pairing.

**Reasons To Accept:**

1.	This work is simple but effective. The paper is easy to follow, and the contributions are clear.
2.	The experimental result on multiple datasets shows that the method is both efficient in time and space.
3.	The analysis of strength and limitations is convincing.


**Reasons To Reject:**

1.	The time cost with different k should be presented, as “a larger k can lead to a better performance for the S-strategy.”
2.	The methods of comparison are a bit outdated. Please include some recent works.



**Reproducibility:**

5: Could easily reproduce the results.

**Reviewer Confidence:**

4: Quite sure. I tried to check the important points carefully. It's unlikely, though conceivable, that I missed something that should affect my ratings.

---

> ### Author Rebuttal · Authors · 2023-08-29
>
> We sincerely appreciate your insightful comments.
>
> **Re: "The time cost with different k should be presented"**
>
> Thank you for your suggestion. Following this, we have added experiments to discuss the time cost with different k. It is shown that MixEncoder consumes more time as k increases. Nevertheless, the difference in timing between Dual-BERT and MixEncoder is rather minimal, whereas Cross-BERT is significantly slower by two orders of magnitude. In our revised version, we will provide additional analysis to elaborate on these findings in Section 5.2 (Ablation Study).
>
> Table: Time to evaluate 100 queries with 1k candidates with the last layer performing the simplified cross-attention.
>
> |       Model       | Time(ms) |
> | :---------------: | :------: |
> |     Dual-BERT     |   7.2    |
> |    Cross-BERT     |  949.4   |
> | MixEncoder (k=1)  |   8.4    |
> | MixEncoder (k=2)  |   9.1    |
> | MixEncoder (k=3)  |   10.0   |
> | MixEncoder (k=4)  |   11.5   |
> | MixEncoder (k=10) |   24.3   |
>
> **Re: "The methods of comparison are a bit outdated. Please include some recent works."**
>
> Thank you for your valuable suggestions. We only considered the methods that improve Dual-BERT by modifying the model architecture before. Following your suggestion, we have added one orthogonal method VIRT [1] which leverages knowledge distillation to improve representation-based text matching. The table below briefly reports the results of this method in the first row. In the revised version, we plan to extend Table 2 within our paper and provide a more comprehensive analysis in Section 5.1.
>
> | Model        | MNLI (acc)   | Ubuntu (MRR) | DSTC7 (MRR)  | MS MARCO (MRR) | Speedup |
> | ------------ | ------------ | ------------ | ------------ | -------------- | ------- |
> | VIRT         | $78.3_{0.3}$ | $89.4_{0.2}$ | $74.8_{0.4}$ | $32.3$         | 40.5x   |
> | MixEncoder-a | $77.5_{0.4}$ | $89.4_{0.1}$ | $74.9_{0.2}$ | $32.0$         | 113x    |
> | MixEncoder-b | $77.8_{0.2}$ | $89.5_{0.1}$ | $75.8_{0.5}$ | $32.5$         | 89.6x   |
>
> [1] [VIRT: Improving Representation-based Text Matching via Virtual Interaction](https://aclanthology.org/2022.emnlp-main.59) (Li et al., EMNLP 2022)

---

### Official Review · Reviewer_7S85 · 2023-08-04

**Soundness:** 3

**Excitement:**

3: Ambivalent: It has merits (e.g., it reports state-of-the-art results, the idea is nice), but there are key weaknesses (e.g., it describes incremental work), and it can significantly benefit from another round of revision. However, I won't object to accepting it if my co-reviewers champion it.

**Paper Topic And Main Contributions:**

In this paper, authors propose a lightweight cross-attention mechanism for sentence matching to preserve the performance and the inference speed of the model simultaneously. Firstly, it attempts to encode candidates into k (k<<l) vectors. When inference, this paper integrates a cross-attention module into some intermediate layers. Experiments on four datasets prove the advantage of the proposed method.

**Reasons To Accept:**

1. The idea of this work is simple but effective.

**Reasons To Reject:**

1. The performance of the model is unstable. On some datasets such as MNLI, Ubuntu, and DSTC7, this work achieves good performance. However, on MS MARCO, it's performance even weaker than the Dual-BERT.

**Reproducibility:**

4: Could mostly reproduce the results, but there may be some variation because of sample variance or minor variations in their interpretation of the protocol or method.

**Reviewer Confidence:**

4: Quite sure. I tried to check the important points carefully. It's unlikely, though conceivable, that I missed something that should affect my ratings.

---

> ### Author Rebuttal · Authors · 2023-08-29
>
> Thank you for your valuable feedback.
>
> **Re: "The performance of the model is unstable."**
>
> In our experiments, we have explored three variants of MixEncoder, namely MixEncoder-a, -b, and -c.  Our experiments on the MS MARCO dataset show that both MixEncoder-a and MixEncoder-b exhibit slight improvements over Dual-BERT in terms of R1@1000,  while MixEncoder--c lags behind Dual-BERT. We have discussed this observation in section A.5 and have provided a potential explanation for these results.
>
> It is also worth mentioning that the baselines considered in our study have not been collectively evaluated on all four datasets (MNLI, Ubuntu, DSTC7, MS MARCO) in prior research. Specifically, Poly-encoder is designed for the Ubuntu and DSTC7 datasets, ColBERT is proposed for MS MARCO, and Deformer is evaluated on MNLI and other NLU tasks.  Given the overall effectiveness of MixEncoder across these four tasks, we believe that MixEncoder represents a general strategy for representation-based matching tasks.

---

### Meta-Review · Area_Chair_DTo3 · 2023-09-23

**Recommendation:** 5

**Metareview:**

All the reviewers have agreed that the idea in this paper is simple yet effective.

However, all the reviewers have concerns on whether the results are convincing enough or not. During the rebuttal period, the authors have addressed some of these concerns.

Generally, the merits of this paper outweigh its flaws.

---

### Decision · Program_Chairs · 2023-10-07

**Decision:**

Accept-Main

**Comment:**

All the reviewers have agreed that the idea in this paper is simple yet effective.

However, all the reviewers have concerns on whether the results are convincing enough or not. During the rebuttal period, the authors have addressed some of these concerns.

Generally, the merits of this paper outweigh its flaws.